# Vestibular Schwannoma Volume and Tumor Growth Correlates with Macrophage Marker Expression

**DOI:** 10.3390/cancers14184429

**Published:** 2022-09-12

**Authors:** Sandra Leisz, Clara Helene Klause, Tania Vital Dos Santos, Pia Haenel, Maximilian Scheer, Sebastian Simmermacher, Christian Mawrin, Christian Strauss, Christian Scheller, Stefan Rampp

**Affiliations:** 1Department of Neurosurgery, Medical Faculty, Martin Luther University Halle-Wittenberg, Ernst-Grube-Str. 40, 06120 Halle (Saale), Germany; 2Department of Neuropathology, Otto von Guericke University Magdeburg, Leipziger Str. 44, 39120 Magdeburg, Germany; 3Department of Neurosurgery, University Hospital Erlangen, Schwabachanlage 6, 91054 Erlangen, Germany

**Keywords:** vestibular schwannoma, tumor growth, tumor volume, macrophage infiltration, CD68, CD163, TAM

## Abstract

**Simple Summary:**

The variable growth behavior of vestibular schwannomas (VS) makes therapy decisions very difficult. These benign tumors, which originate from the eighth cranial nerve, partly show a very slow growth rate over many years. Nevertheless, VS can lead to severe symptoms such as hearing loss and dizziness within a short time due to their increase in size. Despite numerous preliminary studies, no apparent influencing factor on size progression could be found so far. In our study, we consider the influence of growth factors and macrophage markers on the volume and growth rate of VS. While growth factors show no effect on tumor growth, higher expression of macrophage markers indicates an infiltration of macrophages. They may thus enhance the growth of VS and therefore represent a potential therapeutic target.

**Abstract:**

Vestibular schwannoma is the most common benign tumor of the cerebellopontine angle and originates from Schwann cells surrounding the vestibulocochlear nerve. Since the size of the VS varies widely, affected patients suffer from symptoms of varying severity. It is often difficult to determine the optimal time for therapy, due to the unpredictability of the growth rate. Despite many investigations on influencing factors, no mechanism responsible for the increase in the growth rate of certain VS has been identified so far. Therefore, the present study investigates the influence of the seven markers: Ki-67, cyclooxygenase 2 (COX2), vascular endothelial growth factor (VEGF), macrophage colony-stimulating factor (M-CSF), granulocyte-macrophage colony-stimulating factor (GM-CSF), CD163, and CD68 on tumor progression and tumor size in a cohort of 173 VS. The markers were determined by quantitative PCR and correlated with tumor volume and VS growth rate. The analysis showed a significantly negative correlation of the Ki-67, COX2, and VEGF on tumor volume. Moreover, with a higher volume of VS, the expression of the macrophage markers CD68, CD163, and GM-CSF increased significantly. Our results suggest that the increase in VS size is not primarily due to Schwann cell growth but to an infiltration of macrophages. This may have an impact on non-invasive therapy to preserve the hearing function of affected patients.

## 1. Introduction

VS are benign tumors originating from Schwann cells surrounding the eighth cranial nerve. Limited research has been carried out on the development and progression of this tumor. Most VS are associated with a loss of function mutation or promoter methylation of the neurofibromatosis type 2 (NF2) gene [1,2]. Six percent of all cranial tumors are VS and comprise 80% of the tumors in the cerebellopontine angle. Within the last few years, the incidence of VS has increased due to better diagnostic methods such as contrast MRI [3]. Patients with VS do often have unilateral hearing loss. Additional symptoms are tinnitus, vertigo, trigeminal complaints, and facial nerve affection due to the proximity of the tumor to cranial nerves [4,5,6].

The size and growth of VS vary widely. Some VS grow very expansive but not invasive within a few months; others stagnate in growth for several years. While large and small symptomatic tumors are usually treated surgically or irradiated, the growth of smaller asymptomatic tumors is monitored regularly by MRI (wait and scan) [7]. Currently, there is no approved pharmacological therapy for the treatment of VS. Paldor et al. reported that neither sex nor patient age has an impact on VS growth rate. The molecular determinants of progressive or stagnant VS are currently unknown. This makes the prediction of progression and an appropriate point in time for intervention difficult [8]. Since surgical removal carries the risk of postoperative hearing loss, facial nerve palsy, and headache [9], more accurate growth prediction could avoid unnecessary surgical removal, thus reducing the risk of nerve injury. This could lead to longer maintenance of hearing function and thus significantly improve the quality of life of affected patients.

In recent years, several factors have been investigated, which might influence the growth rate of VS. Vascular endothelial growth factor (VEGF) correlated positively with tumor growth rate in a study of 27 patients [10]. 

CD68 and CD163 are two important macrophage markers. While CD68 as a transmembrane glycoprotein is present on the surface of all macrophages, the endocytic receptor CD163 was found mainly expressed on M2 macrophages, which have anti-inflammatory, proto-oncogenic effects [11,12]. M1 macrophages have inflammatory non-oncogenic effects [13]. De Vries et al. examined a correlation between macrophage colony-stimulating factor (M-CSF) and CD163. The expression of both factors was higher in fast-growing VS. [14]. In addition, the authors found a positive correlation between CD68 and tumor size [15]. No preliminary studies have yet been performed on VS on the granulocyte-macrophage colony-stimulating factor (GM-CSF), which can have anti- or pro-tumorigenic effects depending on its expression and tumor immune microenvironment [16,17].

Other previous studies identified Ki-67, a protein only expressed by mitotic cells, and the enzyme cyclooxygenase 2 (COX2) as influencing factors. The VS tumor volume increased with a high expression of Ki-67 [18] as well as COX2 [19]. 

The previous studies mostly investigated only single factors on VS tumor size or growth rate using immunohistochemistry (IHC). The present work investigated the influence of the different factors VEGF, Ki-67, COX2, M-CSF, GM-CSF, CD68, and CD163 on tumor volume and growth as well as on clinical parameters, such as age and hearing, in a cohort of 173 patients using mRNA data and IHC. 

## 2. Materials and Methods

### 2.1. Study Design

Between 2012 and 2021, 208 VS were treated by surgery in the authors’ institution. A total of 35 patients with irradiated VS, recurrence, hereditary neurofibromatosis, and VS from which not enough RNA was isolated for qPCR were excluded. This resulted in a final number of 173 included patients. All participants had a single VS, were adult, had at least one preoperative MRI image, and received microsurgical treatment. Clinical data on age, sex, Koos grade, and hearing class were obtained from the patient’s medical record [20]. The RNA of these 173 VS was used for marker expression analysis by qPCR. A total of 22 patients with MRI images older than 6 months before surgery and those with a slice thickness greater than 2.5 mm were excluded, resulting in 151 patients being part of tumor volume analysis. A total of 77 patients had only one preoperative MRI image and were therefore not suitable for growth rate analysis, which was performed in the remaining 74 patients with multiple preoperative MRI images (Figure 1). The ethics committee of the Medical Faculty of Martin Luther University Halle-Wittenberg (approval number 2020-122) approved the study. Written informed consent was obtained from all patients.

Preoperatively, each patient’s hearing class was determined according to the American Academy of Otolaryngology-Head and Neck Surgery (AAO-HNS) hearing classification, extended by a “D surditas” (DS) category for complete deafness. In addition, preoperative MRI images were used to determine tumor volume by volumetry, and a sample (30–50 mg) was taken during surgical removal to quantify tumor markers.

### 2.2. Tumor Volumetry

One MRI series each with a slice thickness between 0.35 and 2.50 mm was used to analyze the preoperative tumor volume (cm^3^). Only images acquired no more than 6 months prior to surgery were used. Volumetry was performed using the software Brainlab Origin Server 1.1 iPlan Net, version 3.7.0.64 (Brainlab, Munich, Germany). After selecting suitable image sequences (T2-weighted or T1-weighted images, contrasted with gadolinium), the Smart Brush function was used to measure the volume. Growth rate was determined in participants with more than one preoperative MRI image. This involved volumetry of all preoperative images and subsequent calculation of the growth rate in cubic centimeters per year. The minimum interval between preoperative images was 6 months. Growth rate (cm^3^ per year) was calculated in patients with multiple preoperative MRI images using the formula:$$\frac{tumor\;volume\;MRI\;2-tumor\;volume\;MRI\;1}{\left(\Delta \;days\;MRI\;1-2/365\right)}$$

### 2.3. RNA Isolation and Reverse Transcription

A 30–50 mg tumor sample obtained during surgery was used to isolate RNA. A 10-fold volume AllProtect (Qiagen, Hilden, Germany) solution was added to the sample immediately after surgery. The AllPrep DNA/RNA/Protein Mini kit (Qiagen, Hilden, Germany) was used for isolation. Appropriate amount of buffer RLT supplemented with β—mercaptoethanol (1:100, Roth, Karlsruhe, Germany) was added and the sample was disrupted using TissueLyser LT (Qiagen, Hilden, Germany) for 2 min at 50 Hz with stainless steel beads (5 mm, Qiagen, Hilden, Germany). After centrifugation (3 min, 10.000 rpm), supernatant was used for RNA isolation according to the manufacturer’s instructions. Finally, the RNA was eluted in 30 µL of RNase-free water. RNA measurement was performed with the Tecan M200 (Tecan, Männedorf, Schweiz) at wavelengths of 260, 280 and 310 nm. The reverse transcription was carried out according to the instructions of the RevertAid First Strand cDNA Synthesis kit (ThermoFisher Scientific, Waltham, MA, USA) using oligo(dT)_18_-primers. The concentration of cDNA was adjusted to 20 ng/µL. The RNA was incubated for 10 min at 70 °C in the thermocycler PEQLAB primus 25 advanced (VWR, Avantor, Radnor, PA, USA). After addition of the master mix, transcription was performed for 60 min at 42 °C and the reaction was terminated by 5 min at 70 °C.

### 2.4. Quantitative Real-Time PCR

Quantitative real-time polymerase chain reactions (PCR) were performed for determination of the various tumor markers. An amount of 1 µL of the cDNA obtained from the tumor sample at a concentration of 20 ng/µL was used. Two-step PCR was performed using 10 µL of Platinum SYBR Green qPCR SuperMix-UDG (Invitrogen, ThermoFisher Scientific, Waltham, MA, USA) and 0.5 µL each of the corresponding forward and reverse primers (Invitrogen, ThermoFisher Scientific, Waltham, MA, USA, 100 µmol) (Table 1) resulting in a total volume of 20 µL in Rotor-Gene Q (Qiagen, Hilden, Germany). Marker expression was normalized to expression of glyceraldehyde-3-phosphate dehydrogenase (GAPDH). Analysis of the expression data and creation of the heat map was carried out using Prism 9 (GraphPad Software, San Diego, CA, USA).

### 2.5. Immunohistochemistry

Immunohistochemistry (IHC) was performed with specific antibodies against CD68, CD163, COX2, and Ki-67 (Table 2) as previously described in detail [21]. 

### 2.6. Statistical Analysis

The statistical analysis was performed in subgroups according to available data: marker expression analysis in all 173 patients, tumor volume analysis in 151 patients, and growth rate analysis in 74 patients. 

In a first exploratory analysis, correlations between markers as well as clinical variables were evaluated. A classification of the correlation coefficient was used for data interpretation [22]. Correlations in the subgroup with available growth rate data were calculated separately. As AAO-HNS hearing classes are ordinally scaled and the remaining parameters had non-normal distributions, Spearman’s rank correlations were consistently used for all pairings. Due to the exploratory nature of this step, results were not corrected for multiple comparisons. 

Significant correlations with tumor size or growth rates were further investigated using regression analysis. Significant markers or clinical variables then served as predictors whereas tumor size and growth rate separately served as responses. Due to skewed, non-normal distributions, parameters were log-transformed before inputting into the regression model.

To further illustrate the contrast between slow- and fast-growing VS, growth rates were divided into two groups: <0.1 cm^3^/year and >1 cm^3^/year. Cases with growth rates between these ranges were excluded from this analysis step. The two groups were compared using a non-parametric Wilcoxon signed-rank test. Results were not corrected for multiple comparisons. All analysis steps were calculated using R version 4.0.5 [23].

## 3. Results

### 3.1. Correlation Analysis of Markers Investigated and Tumor Volume

In a correlation analysis of investigated markers, tumor volume and clinical data of 173 patients were included. The cohort included 71 male and 102 female patients with a mean age of 51 years (Table 3). The overall mean tumor volume was 3.3 cm^3^ (0.1–37.0 cm^3^) and the median tumor volume was 1.2 cm^3^. The study included a few small VS (Koos 1, n = 11). Most VS corresponded to Koos grade 2 (n = 63) or Koos grade 3 (n = 61). Displacing growing VS were detected in 38 patients (Koos 4). While the majority of participants had little or no limitation in their hearing function (AAO-HNS A, B), 59 patients suffered from advanced hearing loss (AAO-HNS C, D). In 10 of the 173 participants, complete hearing loss was detected preoperatively (AAO-HNS DS).

For evaluation, clinical data were correlated with the quantitative determinations of the markers using qPCR. An overview of the mRNA marker expression is shown as a heat map (Appendix A). Correlation analysis of the markers COX2, VEGF, Ki-67, CD163, CD68, GM-CSF and M-CSF with age, tumor volume, and hearing class were performed using Spearman’s rank correlation coefficient (Figure 2). The angiogenesis promoting factor VEGF correlated moderately positively with the growth factor Ki-67 (r = 0.42, *p* < 0.0001) and weakly positively with COX2 (r = 0.19, *p* = 0.01), as well as with M-CSF (r = 0.15, *p* = 0.048) in VS. The macrophage differentiation factor M-CSF additionally correlated weakly positively with Ki-67 (r = 0.28, *p* < 0.0001) and moderately positively with the two macrophage antigens CD68 (r = 0.46, *p* < 0.0001) and CD163 (0.52, *p* < 0.0001). In VS, high expression of CD68 led to a strong increase in CD163 expression (r = 0.69, *p* < 0.0001). Only CD68 and COX2 correlated weakly negatively with each other (r = −0.19, *p* = 0.01). The analysis of GM-CSF expression showed significant negative correlations with VEGF (r = −0.32, *p* < 0.0001) and Ki-67 (r = −0.34, *p* < 0.0001). The higher the tumor volume of a VS, the lower the expression of COX2 (r = −0.39, *p* < 0.0001), VEGF (r = −0.21, *p* = 0.009), and Ki-67 (r = −0.34, *p* < 0.0001). With progressive expansion of the VS and thus increasing tumor volume, the expression of the macrophage markers CD68 (r = 0.28, *p* < 0.0001) and CD163 (r = 0.19, *p* = 0.02) increased weakly. The correlation of the clinical parameters age and hearing class showed a moderate positive correlation (r = 0.4, *p* < 0.0001). In addition, hearing function correlated weakly positively with CD163 (r = 0.15, *p* = 0.049) (Figure 2). Further statistically significant correlations could not be identified. The Supplementary contains the correlation analysis, supplemented by the Koos classification (Appendix A).

### 3.2. Regression Analysis of Significant Markers and Tumor Volume

For a more detailed examination of the correlations, regression analysis with the response variable tumor volume was performed. When examining the significant markers from the correlation analysis, three markers were identified that had a significant effect on tumor volume even without the influence of the other markers. COX2 (*p* = 0.003) and Ki-67 (*p* = 0.0002) showed a statistically significant negative association with tumor volume, as previously presented in the correlation analysis. CD163 expression increased in larger VS even without the involvement of the other tumor markers (*p* = 0.08) (Figure 3).

### 3.3. Correlation Analysis of Markers Investigated and Tumor Growth Rate

For correlation analysis of VS growth rate, all patients with multiple preoperative MRI images were included. Thus, the analysis involved 74 patients, of whom 32 are male and 42 are female. The mean tumor volume was 2.44 cm^3^ (0.1–18.8 cm^3^) and the median tumor volume was 0.8 cm^3^. The mean growth rate per year was 1.0 cm^3^ (0.01–17.5 cm^3^) and the median growth rate was 0.3 cm^3^ (Table 4).

The correlation analysis of the growth rate including the markers was investigated with Spearman’s rank correlation coefficient. The growth rate correlated very strongly with tumor volume (r = 0.86, *p* < 0.0001) in VS. Furthermore, hearing class worsened weakly with a higher growth rate (r = 0.3, *p* = 0.008). In addition to the correlation of hearing class with age (r = 0.34, *p* = 0.002) and growth rate, this analysis showed weak positive correlations with tumor volume (r = 0.3, *p* = 0.011). Patients with VS with higher expression of the growth factors COX2 (r = −0.35, *p* = 0.002) and VEGF (r = −0.24, *p* = 0.038) had a lower hearing class. Patient age correlated weakly negatively with COX2 expression (r = −0.23, *p* = 0.049). COX2 also showed a correlation with growth rate (r = −0.23, *p* = 0.049). Although the correlation between CD68 and growth rate was not significant, a weak increase in CD68 expression in rapidly growing VS was detectable. Considering only the patients with growth rate, GM-CSF also correlated positively with tumor volume (r = 0.26, *p* = 0.045) (Figure 4). Furthermore, a correlation analysis including the Koos classification was performed (Appendix A).

### 3.4. Regression Analysis of Tumor Volume and Tumor Growth Rate

The very strong correlation of tumor volume with the growth rate of VS could be demonstrated in the subsequently performed regression analysis. The response variable growth rate is significantly higher in large VS without the influence of other factors (*p* = 0.003). The growth rate tended to increase with higher tumor volume (Figure 5).

### 3.5. Boxplot Analysis in Fast- and Slow-Growing VS

To further illustrate the influence of VS growth rate, patients were divided into two groups. VS with a growth rate < 0.1 cm^3^ per year correspond to slow-growing VS and those with a growth rate > 1 cm^3^ per year correspond to fast-growing ones. The boxplot analysis included 16 patients with slow-growing VS with a mean age of 49 years and a mean tumor volume of 0.3 cm^3^. The mean growth rate of slow-growing VS per year was 0.05 cm^3^. The mean age of the 17 patients with fast-growing VS was 57 years. Fast-growing VS had a mean tumor volume of 7.2 cm^3^ and a mean growth rate of 3.5 cm^3^ per year (Table 5).

A comparison between slow- vs. fast-growing VS revealed five statistically significant differentially expressed markers. Slow-growing VS were presented with a significantly higher expression of COX2 than fast-growing VS (*p* = 0.023). The median COX2 expression of slow-growing schwannomas was nearly four times as high as the ones of the fast-growing schwannomas (median slow-growing = 0.96; median fast-growing = 0.27) (Figure 6).

In contrast, the expression of the macrophage marker CD68 was significantly higher in VS with a growth rate > 1 cm^3^ per year (median = 1.81) than in those with a growth rate < 0.1 cm^3^ per year (median = 0.99) (*p* = 0.027). 

Furthermore, the comparison of fast- and slow-growing VS showed significant correlations when considering the clinical parameters of tumor volume and hearing class. While slow-growing VS showed a median tumor volume of 0.3 cm^3^, fast-growing VS had a median tumor volume of 5.3 cm^3^ (*p* < 0.0001). Patients with slow-growing VS had on average moderately impaired hearing function. The hearing performance of patients with fast-growing VS was one hearing class worse (*p* = 0.013).

Although CD163 expression did not show a statistically significant correlation when comparing the two groups of fast- and slow-growth VS, its expression was higher in fast-growing VS than in slow-growing VS. The other markers determined in the study did not show any significant correlations between the two groups of fast- and slow-growing VS.

### 3.6. IHC Analyses of CD68, CD163, Ki-67, and COX2

For further consideration of our results on the mRNA level, an exemplary investigation of the significant markers at the protein level using IHC was performed. For this, we randomly selected three VS samples each with a tumor volume < 0.5 cm^3^, three VS each with a tumor volume > 5 cm^3^, three patients each with a fast-growing VS (>1 cm^3^/year) and three with a slow-growing VS (<0.1 cm^3^/year). We obtained overview images at 40× magnification (Appendix A) and images at 200× magnification of the four markers, which correlated significantly at mRNA level (Figure 7). The results show that the protein levels of both macrophage markers CD68 (Figure 7a) and CD163 (Figure 7b) are higher in the three larger VS than in the smaller VS. In contrast, a lower protein amount of Ki-67 (Figure 7c) and COX2 (Figure 7d) could be detected in the large VS by IHC.

In addition, IHC analyses were performed on the two markers, which were detected to be significantly differentially expressed with regard to the different growth rates (boxplot analyses, Figure 6). Abundant CD68 expression was detected in the fast-growing VS. (Figure 8a). In addition, protein expression of COX2 was decreased in the fast-growing VS. (Figure 8b). Furthermore, overview images were captured at 40× magnification (Appendix A).

## 4. Discussion

In recent years, there has been a surge of interest in identifying a possible factor influencing the growth of sporadic VS. Hong et al. suggested that one explanation for a higher growth rate is the increased expression of COX2 in rapidly growing VS. [24]. The authors examined COX2 expression in 15 sporadic VS by IHC and found significantly higher expression in fast-growing VS. In addition, a positive correlation between COX2 expression and tumor extension (Hannover classification) in 1048 VS was detected by IHC [19]. However, no increase in COX2 expression was found with an increasing volumetric growth rate [25]. Treatment with COX2 inhibitors did not affect COX2 expression and VS- growth [19]. In a retrospective study of 347 patients, regular MRI scans showed a difference in VS growth in patients taking the pan-COX inhibitor aspirin regularly versus patients not taking aspirin. Kandathil et al. therefore suggested a possible therapeutic role of COX2 inhibitors to prevent VS growth [26,27]. An in vitro study of VS primary cultures also demonstrated an inhibitory effect of aspirin on VS proliferation, which was not confirmed in healthy Schwann cells [28]. 

A correlation between increased COX2 expression and tumor progression is suspected in various malignant tumors [29]. In breast and colorectal cancer, such a correlation has been identified. [30,31]. However, VS is a benign tumor that may have different growth characteristics compared to malignant tumors. Our study showed an opposite effect on tumor volume. The expression of COX2 was decreased in both large VS and fast-growing VS. This result was also confirmed by regression analysis and exemplary IHC analysis. In contrast to previous studies, our results showed an opposite effect of the prostaglandin-producing enzyme COX2 on the progression of VS. 

In many prior studies, cell mitosis was detected by the diagnostic cell proliferation marker Ki-67, which was associated with increased COX2 expression but showed no effect on greater tumor volume or higher growth rate [32,33]. Koutsimpelas et al. examined a positive correlation between proliferation index and tumor diameter in 182 sporadic VS [18]. In contrast to this, analysis of 747 VS showed increasing Ki-67 expression in smaller tumors and VS with a higher growth rate [25]. In contrast to the previous study, our study showed lower expression of Ki-67 in large VS and in fast-growing VS. Abundant Ki-67 expression was also detected in the IHC sections of the smaller VS. Therefore, it is likely that the proliferation of Schwann cells does not play a prominent role in the growth of VS. 

The angiogenesis-promoting factor VEGF represents a key factor in the angiogenesis of malignant tumors, such as lung carcinoma [34]. As a mediator of angiogenesis in tumors, it promotes new vessel growth to supply oxygen and nutrients [35]. Nevertheless, Brieger et al. found no evidence of VEGF influencing VS size progression [36]. Other studies showed that the growth rate correlated positively with VEGF expression, being significantly higher in VS than in healthy nerves [10,37]. In contrast to these studies, we showed that VEGF expression was decreased with increasing tumor volume. Although angiogenesis plays a crucial role in the progression of various tumors [35], this does not seem to apply to the VS. This supports our hypothesis that tumor growth is not exclusively due to the proliferation of Schwann cells, which are dependent on blood nutrient supply.

Studies increasingly show that the tumor microenvironment influences the progression of tumors [38]. An influence of this milieu was also discussed for VS. Hannan et al. assume an inflammatory process that promotes growth [39]. Analyzing 19 sporadic VS by positron emission tomography (PET) scan showed increased inflammatory activity in growing VS [40]. Although COX2 is a pro-inflammatory enzyme, the increase in inflammatory processes may have other causes. Therefore, immune cell infiltration and tumor microenvironment of VS need to be better elucidated to understand tumor development and to find suitable targets for drug therapy [40,41,42]. In particular, infiltration of macrophages has been the subject of numerous preliminary studies. 

Two main phenotypes of polarized macrophages are characterized. Whereas M1 macrophages are mainly involved in inflammatory non-oncogenic processes, M2 macrophages have anti-inflammatory, proto-oncogenic functions [13]. Tumor-associated macrophages (TAMs), which play a role in the development of various tumor entities, are similar to M2 macrophages [43] and express the macrophage antigen CD163 [44]. De Vries et al. demonstrated higher expression of the macrophage marker CD68 in large, rapidly growing VS in 67 sporadic VS [15]. Perry et al. indicated the dominance of CD163 negative M1 macrophages in 46 large sporadic VS. The most likely explanation is the hypothesis that M1/M2 polarization in the VS is more complex, but the interaction of TAMs is crucial for progression [45]. Nonetheless, the importance of M2 macrophages has been already demonstrated in another study. An IHC study of 923 VS demonstrated higher expression of CD68 and CD163 in larger tumors, suggesting an influence of M2 macrophages in large vs., confirming our results [46,47]. However, in contrast to our study, no correlation with a higher growth rate could be demonstrated. In the included 173 VS, we detected increased macrophage marker expression in fast-growing VS. The expression of CD68, as well as CD163, was higher in larger VS, indicating infiltration of macrophages. This may reflect the increasing involvement of M2 macrophages in larger VS. In addition, regression analysis showed that CD163 was increasingly expressed in larger VS even without the influence of CD68.

De Vries et al. compared CD163 expression in 10 fast- and 10 slow-growing VS and demonstrated significantly higher involvement of M2 macrophages in fast-growing VS [46]. We also examined the differences between these two groups. The expression of CD68 was significantly higher in the 17 fast-growing VS than in the 16 slow-growing VS. This highlights the strong involvement of macrophages in growth progression. Although no significant difference in CD163 expression was detected in our study, the non-parametric Wilcoxon signed-rank test shown in the boxplots clearly indicates greater involvement of M2 macrophages in the rapidly growing VS. After including an inflammatory score combining the lymphocyte and macrophage expression, Goncalves et al. demonstrated a correlation between slower growth rate and a high inflammatory score [46,47]. This contradicts our results indicating infiltration of M2 macrophages in fast-growing VS, also shown in IHC analyses of fast-growing VS. Our results are corroborated by the fact that TAMs are suspected to promote tumor growth by secreting growth factors, cytokines, and chemokines, such as transforming growth factor-β (TGF-β) and chemokine C-X-C motif ligand (CXCL) [48]. Future studies should investigate the expression of these factors in correlation with TAM involvement. 

The macrophage colony-stimulating factor (M-CSF), which promotes the maintenance of TAMs [49], was quantitatively investigated for the first time on a large cohort of VS in our study. De Vries et al. demonstrated higher M-CSF expression in fast-growing VS. [14]. Our study showed a strong correlation of M-CSF with CD163. This supports the hypothesis that M-CSF maintains the function of TAMs and is involved in the VS growth progression. While M-CSF is expressed during homeostatic conditions, GM-CSF expression can be increased during inflammatory processes [17]. Whether GM-CSF has pro- or anti-tumorigenic effects depends on its expression and tumor microenvironment. In glioblastoma, for example, increased expression of GM-CSF leads to tumor growth [50,51]. So far, no information on the influence of GM-CSF on VS volume increase is known. In addition to various other cells, tumor cells also produce GM-CSF and recruit macrophages [49]. This occurs through increased polarization of M1 to M2 macrophages by GM-CSF [16]. Thus, the detection of higher GM-CSF expression in larger VS could lead to increased macrophage, particularly TAM, recruitment, supporting our hypothesis that macrophage infiltration plays a role in VS volume increase (Figure 9). 

In addition to tumor marker analysis, we also examined one clinical aspect. The most common symptom of patients with sporadic VS is unilateral hearing loss [7]. Therefore, in our study, we examined the correlation of the investigated markers and tumor volume with hearing function. Patel et al. determined a dependence between worse hearing function and larger VS in 230 patients [52]. This is supported by our results. However, we additionally showed the negative influence of a higher growth rate on hearing function. One possible explanation could be that the cochlear nerve has more time to adapt to the conditions caused by volume increase in slow-growing VS rather than in fast-growing VS. With increasing VS volume, hearing function decreases significantly and severely limits the quality of life of affected patients. By demonstrating possible causes for a faster growth rate in our study, we provide the basis for a targeted therapy to reduce the growth rate and preserve hearing function for a longer period.

In summary, our study suggests that the progression of VS is not promoted exclusively due to the proliferation of schwannoma cells. In fact, we show that schwannoma cell proliferation is higher in smaller VS. One possible explanation for the faster growth and larger volume is the infiltration of VS by macrophages, specifically by TAMs. However, it is also possible that increased expression of macrophage markers in large VS initiates a downstream signaling pathway, which could lead to an increase in tumor cells in the VS. In addition, TAMs could play a signaling role in this process. Thus, targeting TAMs offers a potential pharmacological target for the treatment of larger growing VS. There are already several targets and clinical trials for drug therapies focusing on the inhibition of TAM-secreted proteins, reprogramming to anti-tumoral macrophages and inhibition of TAM receptors such as Axl. Bemcentinib, an inhibitor of this receptor, is already the subject of clinical trials [13,53]. In the long term, the personalized pharmacological treatment of VS and thereby the prevention of growth progression should be the focus of further investigations. 

### Limitations

Our study results are limited by the fact that only VS with MRI follow-up images could be included when considering the growth rate. This usually only concerns VS that are initially often rather mild and small at diagnosis, since larger VS are operated on promptly. Therefore, large VS are less represented in the growth rate analysis. Another limitation is the exploratory character of our study and the lack of correction for multiple comparisons in the statistical analysis of our results.

## 5. Conclusions

The expression of growth factors was higher in small, slow-growing VS. In contrast, the macrophage markers CD68 and CD163 showed higher expression in large, rapidly growing VS. Thus, the size progression of VS does not seem to originate from the growth of Schwann cells but from infiltration by macrophages. In particular, TAMs seem to play a major role in this process. Future studies should address the possibility of targeted pharmacological treatment of TAMs in VS.

## Figures and Tables

**Figure 1 cancers-14-04429-f001:**
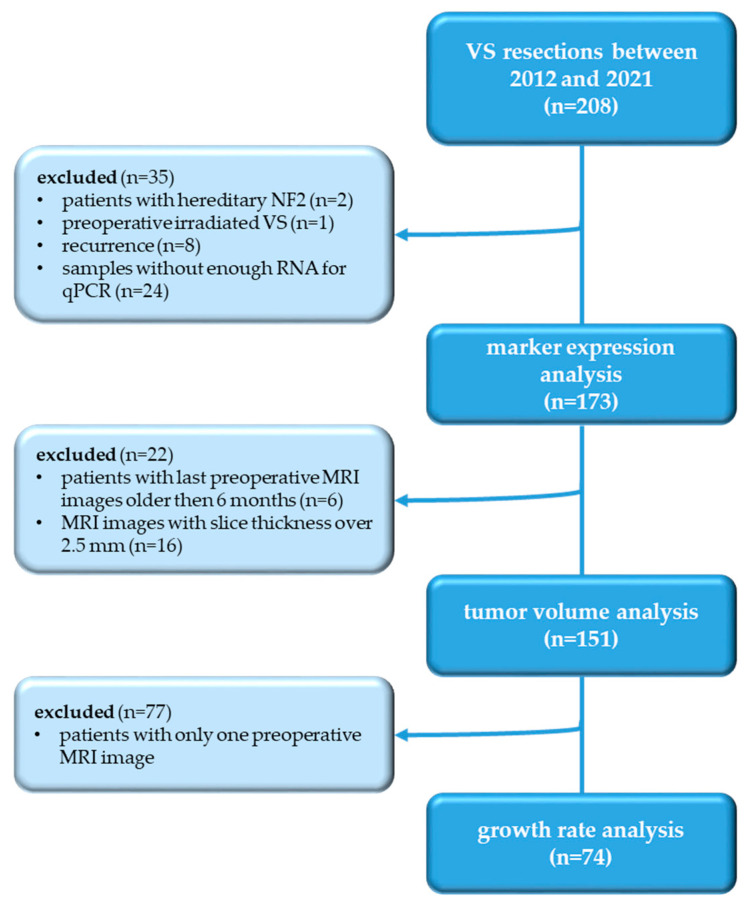
Flow diagram of the study.

**Figure 2 cancers-14-04429-f002:**
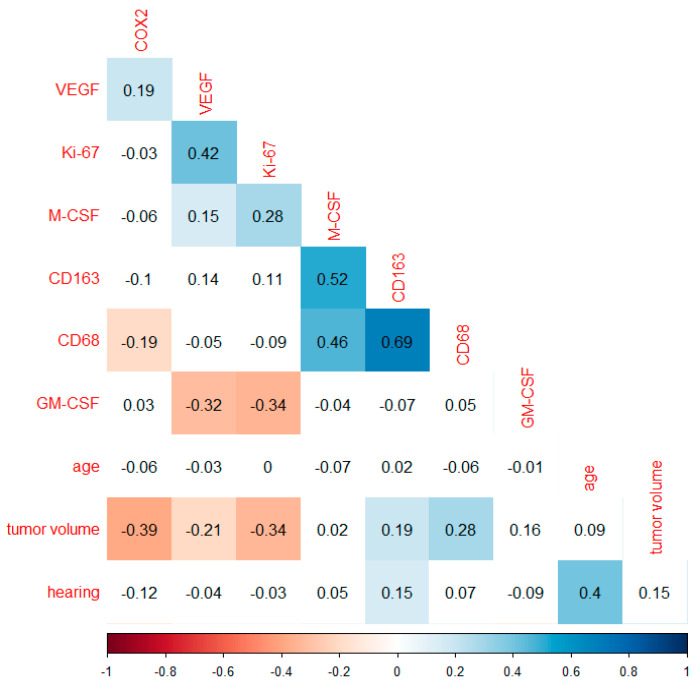
Correlation analysis with Spearman’s rank correlation coefficient. Plot of correlations includes markers investigated, age and hearing class in 173 patients, correlation analysis of markers investigated and tumor volume in 151 patients. The numbers shown correspond to the correlation value r, also reflected by the color shading. Non-significant correlations (*p* ≥ 0.05) are shown with a white background.

**Figure 3 cancers-14-04429-f003:**
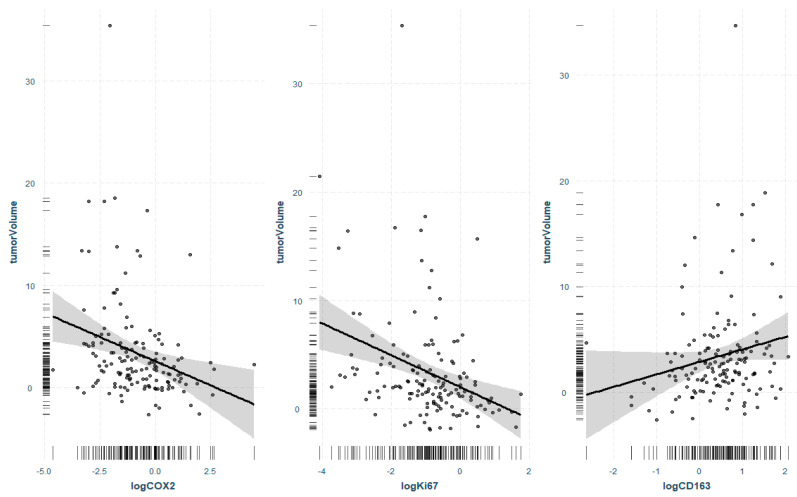
Regression analysis with response variable tumor volume and factors COX2, Ki67, and CD163. Only markers that were significant in correlation analysis and regression were considered. Data are log-transformed due to skewed non-normal distribution and correspondingly displayed on a logarithmic scale. Grey area shows 95% confidence interval, points show partial residuals, the rug lines on both axes show the distribution of the data.

**Figure 4 cancers-14-04429-f004:**
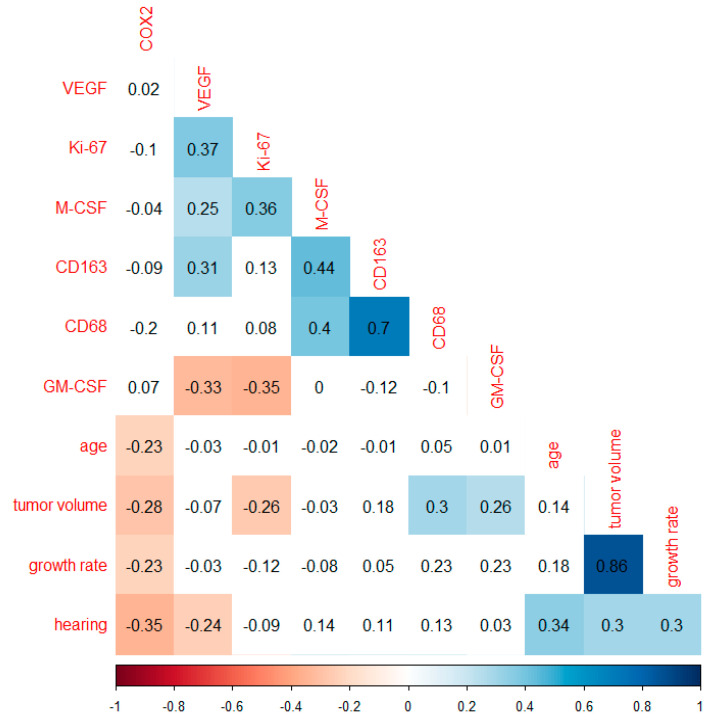
Correlation analysis with Spearman’s rank correlation coefficient in 74 patients with growth rate. Plot of correlations includes markers investigated, tumor volume, growth rate, age, and hearing class. The numbers shown correspond to the correlation value r, also reflected by the color shading. Non-significant correlations (*p* ≥ 0.05) are shown with a white background.

**Figure 5 cancers-14-04429-f005:**
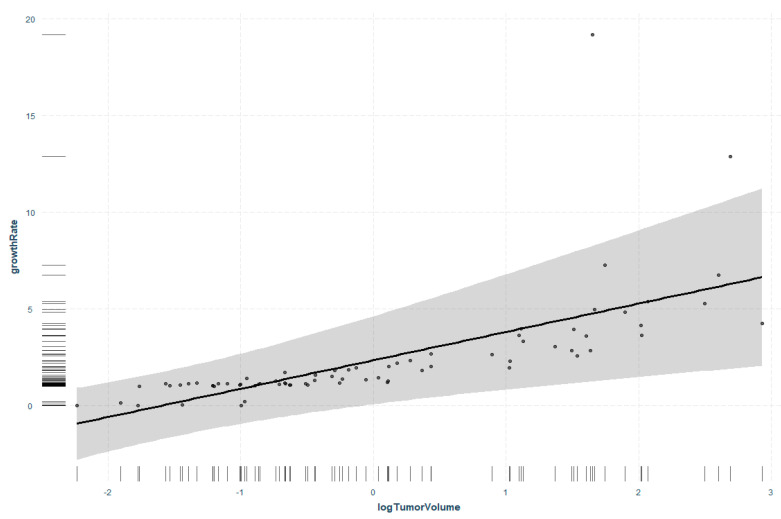
Regression analysis with response variable growth rate and factor tumor volume. Data are log-transformed due to skewed non-normal distribution and correspondingly displayed on a logarithmic scale. Grey area shows 95% confidence interval, points show partial residuals, the rug lines on both axes show the distribution of the data.

**Figure 6 cancers-14-04429-f006:**
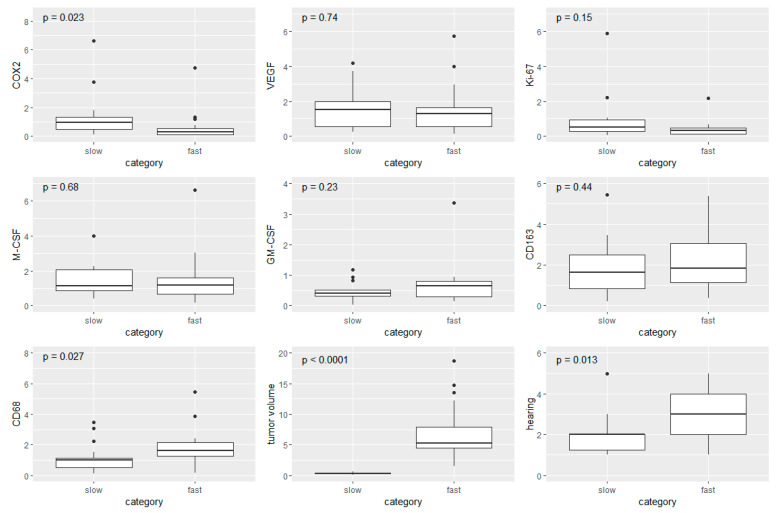
Boxplots of slow- and fast-growing VS. Looking at the seven markers investigated, as well as tumor volume and hearing class, the boxes represent the interquartile range and the crossbar marks the median value. Whiskers illustrate extreme values, excluding outliers, which are outside of 1.5 times the interquartile distances above/below the third/first quartile and depicted as points.

**Figure 7 cancers-14-04429-f007:**
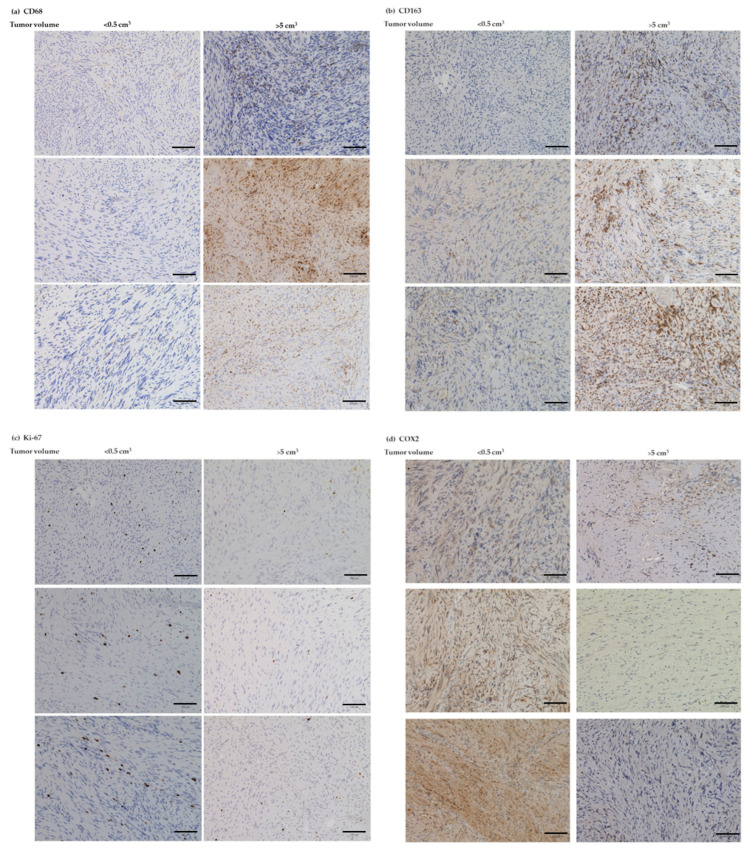
Exemplary IHC analyses of macrophage markers CD68 (**a**) and CD163 (**b**) as well as Ki-67 (**c**) and COX2 (**d**) in VS with tumor volume < 0.5 cm^3^ (left side) and >5 cm^3^ (right side). Bar = 100 µm.

**Figure 8 cancers-14-04429-f008:**
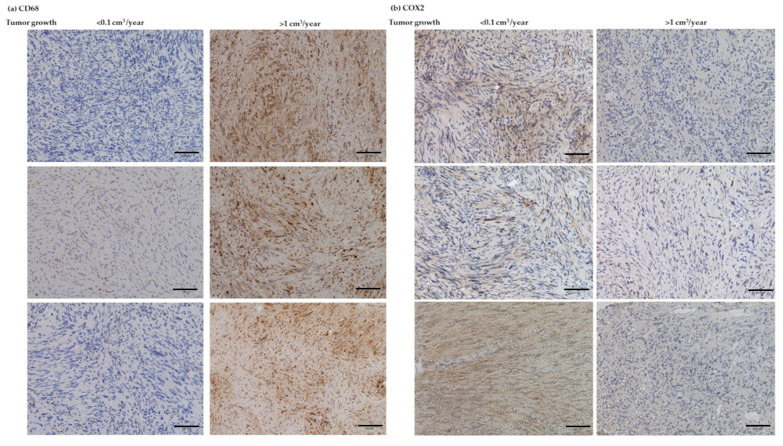
Exemplary IHC analyses of macrophage marker CD68 (**a**) and COX2 (**b**) in VS with growth rate < 0.1 cm^3^/year (left side) and >1 cm^3^/year (right side). Bar = 100 µm.

**Figure 9 cancers-14-04429-f009:**
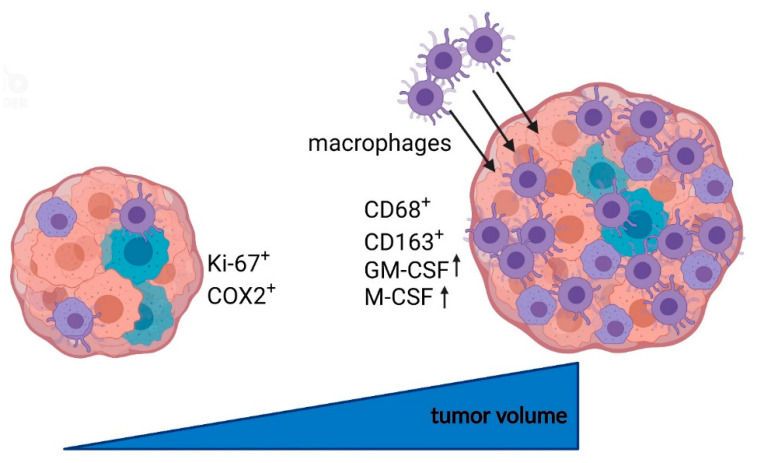
Schematic overview of the putative influence of macrophages on tumor growth in vestibular schwannoma. Smaller tumors showed increased expression of Ki-67 and COX2, whereas large VS were characterized by increased macrophage marker expression (CD68, CD163) and macrophage-associated factors (GM-CSF, M-CSF). These results suggest that macrophages and macrophage-stimulating factors play a crucial role in VS size increase and may represent putative targets for therapeutic and diagnostic markers. Created with BioRender.com.

**Table 1 cancers-14-04429-t001:** Primers used for real-time quantitative PCR.

Gene Name	Oligo Sequence 5′ to 3′ (Forward, Reverse)	Annealing Temperature (°C)	Product Length	Reference Sequence	Species
*CD68*	CGCAGCACAGTGGACATTCT GGATCAGGCCGATGATGAGAG	60	236	NM_001251.3	Homo sapiens
*CD163*	AAAAAGCCACAACAGGTCGC ATGGCCTCCTTTTCCATTCCA	60	322	NM_004244.5	
*COX2*	CCCTTCTGCCTGACACCTTT TTCTGTACTGCGGGTGGAAC	60	203	NM_000963.3	
*Ki-67*	GATCGTCCCAGTGGAAGAGTT ATTGCCTCCTGCTCATGGATT	60	276	NM_002417.5	
*M-CSF*	CCAGAAGGAGGACCAGCAAG CCAAGGGAGAATCCGCTCTC	60	244	NM_000757.6	
*GM-CSF*	AGACACTGCTGCTGAGATGAAT AGGAAGTTTCCGGGGTTGG	60	197	NG_033024.1	
*VEGFA*	AACCATGAACTTTCTGCTGTCTTGG ATCAGGGTACTCCTGGAAGATGTCC	60	205	NM_001171630.1	
*GAPDH*	TCGTGGAAGGACTCATGACC TTCCCGTTCAGCTCAGGGAT	60	172	NM_002046.7	

**Table 2 cancers-14-04429-t002:** Antibodies used for IHC.

Antibody	Source/Isotype	Dilution	Manufacturer
CD68	Mouse IgG	1:4000	Agilent, Santa Clara, CA, USA
CD163	Rabbit IgG	1:500	Invitrogen, ThermoFisher Scientific, Waltham, MA, USA
COX2	Rabbit IgG	1:400	Cell signaling, Danvers, MA, USA
Ki-67	Mouse IgG	Ready to use	Agilent, Santa Clara, CA, USA

**Table 3 cancers-14-04429-t003:** Baseline data for correlation analysis of 173 patients with VS.

Variable	Total (n = 173)
Male Female Mean age (years)	71 102 51 (18–77)
Koos grade 1 2 3 4	11 63 61 38
AAO-HNS ^1^ (hearing class) A B C D DS	56 47 41 18 10
Mean tumor volume (cm^3^) Median tumor volume (cm^3^)	3.3 (0.1–37.0) 1.2

^1^ AAO-HNS: American Academy of Otolaryngology-Head and Neck Surgery.

**Table 4 cancers-14-04429-t004:** Baseline data for correlation analysis of 74 patients with VS with growth rate.

**Variable**	**Total (n = 74)**
Male Female Mean age (years)	32 42 53 (28–77)
Koos grade 1 2 3 4	8 36 17 13
AAO-HNS ^1^ (hearing class) A B C D DS	18 23 19 7 6
Mean tumor volume (cm^3^) Median tumor volume (cm^3^) Mean growth rate (cm^3^/year) Median growth rate (cm^3^/year)	2.44 (0.1–18.8) 0.8 1.0 (0.01–17.5) 0.3

^1^ AAO-HNS: American Academy of Otolaryngology-Head and Neck Surgery.

**Table 5 cancers-14-04429-t005:** Baseline data for boxplot analysis of 16 slow- and 17 fast-growing VS.

**Variable**	**Slow-Growing VS (n = 16)**	**Fast-Growing VS (n = 17)**
Male Female Mean age (years)	6 10 49 (28–70)	9 8 57 (31–77)
Koos grade 1 2 3 4	4 12 0 0	0 0 7 10
AAO-HNS ^1^ (hearing class) A B C D DS	5 8 2 0 1	1 6 4 2 3
Mean tumor volume (cm^3^) Mean growth rate (cm^3^/year)	0.3 (0.1–0.6) 0.05 (0.01–0.1)	7.2 (1.5–18.8) 3.5 (1.0–17.5)

^1^ AAO-HNS: American Academy of Otolaryngology-Head and Neck Surgery.

## Data Availability

The data presented in this study are available on request from the corresponding author.

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
