# Peer review of "Vestibular Schwannoma Volume and Tumor Growth Correlates with Macrophage Marker Expression"

_cancers, 2022, doi:10.3390/cancers14184429_

Round 1
Reviewer 1 Report (Previous Reviewer 1)
The authors thoroughly revised the manuscript as I recommended.
Author Response
Please see the attachment.

Reviewer 2 Report (Previous Reviewer 2)
This paper remains a bit imprecise or even slightly careless at times but has been much improved and is probably OK to publish after minor points below have been addressed. I suggest for their next paper that the authors seek professional help, not so much for their English language use which is OK, but rather for their weak manner of exposition.
I would ask the authors to not use ambiguous phrases like "Macrophages may thus play a role in the growth of VS and therefore represent a potential therapeutic target." That could mean macrophages enhance growth or inhibit growth. If the authors mean to say both then say that
longhand, don't leave readers wondering.
As a personal reading of the research database on GB, and the common non-CNS cancers, macrophages & monocyte lineage cells play mostly a growth enhancing role. Would the authors like to comment on the non-malignant growth characteristics of VS [expansive but not invasive, non-metastatic] and how that might be related to the inverse macrophage-growth relationship in VS compared to the direct relationship in, f. ex. GB ?
First mention in Abstract of what an abbreviation stands for then not repeating this in Text is not typical but might be OK.
Line 24 -...on tumor enhancement, higher expression of macrophage markers indicates an infiltration of immune cells - The authors make a leap from macrophage to immunity. I think we cant do that. Yes, monocyte lineage cells are a critical link in immune responses but they are also active in nonimmune inflammation, in simple trophic and noninflammation related debris clearing i.a. Right ? The authors have done no work implicating immune responses in VS growth or growth inhibition. We gain nothing but unclarity here in extrapolating or shifting terminology from macrophage to immune cells.
Line 24 i recommend not using word enhancement because this word could bring to mind CT or MRI image enhancement.
Line 52 should read "contrast MRI".
Line 61 typo - VS-growth rate
Line 61 either "molecular determinants of " or " differences between "
Line 62 " progression " not progress
Line 71 As a personal preference I recommend not using a decimal, not showing the tenth place, when this adds nothing. F. ex. Table 3 stating 51.1 (18-77) the decimal, .1, adds nothing. Strong communication demands simplifying information without detracting from message. Do the authors really think a tenth of a year is a significant bit of information here ?
Line 77 This is incorrect. G(M)-CSF can also increase MDSC. They can exert thereby an immune and inflammation suppressing effect. The database showing this is huge. Some examples listed below.
Line 80 Is incorrect also. COX2 is not really an inflammatory enzyme. It creates PGE2 but my reading of the literature is that PGE2, like G(M)-CSF, is a "double edged sword".
Re. Fig.3, would the authors think it worthwhile making a single composite score constructed from the three parameters, logCOX2, log Ki67, and log CD163 for X axis entry, with Y axis remaining tumor volume, and seeing if that graph has less spread, tighter correlation ?
Sorry, I don't understand line 303, "Although CD163 showed no statistically significant correlation..." yet then say " its expression was higher in fast-growing VS than in slow-growing VS." ?
Line 303 Although CD163 showed no statistically significant correlation… with what ?
Line 357 typo.
Fig. 7 adds value to the paper. Nice.
Line 349 is so misleading as to be wrong or worse, dishonest. Reckamp et al, ref 28, showed that celecoxib as adjunct might have had growth reduction activity. Also if the authors embark on the subject of celecoxib use in cancer this must be in its own paragraph and the follow up data to their ref 28 MUST be discussed. See numbers 10, 11, 12 below. Furthermore the main point of Reckamp et al ref 28 was that for meaningful COX2 inhibition a daily dose of 1200 mg will be required. The follow on paper of Edelman et al, number 12 below, used 800 mg-day, a dose Reckamp showed was too low for use in cancer treatment.
Note also data on celecoxib 800 mg-day has shown evidence of potential - but equivocal - activity in other cancers, refs 13 to 17 below are examples of recent clinical studies in cancer using celecoxib 800mg.day that also must be included in any discussion of celecoxib in cancer.
Line 357 is confusing, seems to contradict Fig. 6, Fig. 8 and other results. ??
Line 356 Lower case a in aspirin. Rule both in English and in medical English is that proprietary drug name start with upper case, generic drug name starts with lower case. This corresponds with the general English use, proper names start with upper case, otherwise general nouns start with lower case.
Line 373, not grafting. New vessel growth.
Line 458 is a core finding. Emphasize this in Abstract and in Introduction.
========================================================================
1: Kumar A, Taghi Khani A, Sanchez Ortiz A, Swaminathan S. GM-CSF: A Double-Edged Sword in Cancer Immunotherapy. Front Immunol. 2022 Jul 5;13:901277. doi:10.3389/fimmu.2022.901277. PMID: 35865534; PMCID: PMC9294178.
2: Hong IS. Stimulatory versus suppressive effects of GM-CSF on tumor
progression in multiple cancer types. Exp Mol Med. 2016 Jul 1;48(7):e242. doi:10.1038/emm.2016.64. PMID: 27364892; PMCID: PMC4973317.
3: Ruffolo LI, Jackson KM, Kuhlers PC, Dale BS, Figueroa Guilliani NM, Ullman NA, Burchard PR, Qin SS, Juviler PG, Keilson JM, Morrison AB, Georger M, Jewell R, Calvi LM, Nywening TM, O'Dell MR, Hezel AF, De Las Casas L, Lesinski GB, Yeh JJ, Hernandez-Alejandro R, Belt BA, Linehan DC. GM-CSF drives myelopoiesis, recruitment and polarisation of tumour-associated macrophages in cholangiocarcinoma and systemic blockade facilitates antitumour immunity. Gut. 2022 Jul;71(7):1386-1398. doi: 10.1136/gutjnl-2021-324109. Epub 2021 Aug 19.PMID: 34413131; PMCID: PMC8857285.
4: Parmiani G, Castelli C, Pilla L, Santinami M, Colombo MP, Rivoltini L.
Opposite immune functions of GM-CSF administered as vaccine adjuvant in cancer patients. Ann Oncol. 2007 Feb;18(2):226-32. doi:10.1093/annonc/mdl158. Epub2006 Nov 20. PMID: 17116643.
5: Ma N, Liu Q, Hou L, Wang Y, Liu Z. MDSCs are involved in the protumorigenic potentials of GM-CSF in colitis-associated cancer. Int J Immunopathol Pharmacol. 2017 Jun;30(2):152-162. doi: 10.1177/0394632017711055. Epub 2017 May 23. PMID:
28534709; PMCID: PMC5806790.
6: Aliper AM, Frieden-Korovkina VP, Buzdin A, Roumiantsev SA, Zhavoronkov A. A role for G-CSF and GM-CSF in nonmyeloid cancers. Cancer Med. 2014Aug;3(4):737-46. doi: 10.1002/cam4.239. Epub 2014 Apr 2. PMID: 24692240; PMCID:PMC4303143.
7: He K, Liu X, Hoffman RD, Shi RZ, Lv GY, Gao JL. G-CSF/GM-CSF-induced
hematopoietic dysregulation in the progression of solid tumors. FEBS Open Bio. 2022 Jul;12(7):1268-1285. doi: 10.1002/2211-5463.13445. Epub 2022 Jun 9. PMID:35612789; PMCID: PMC9249339.
8: Gutschalk CM, Yanamandra AK, Linde N, Meides A, Depner S, Mueller MM. GM-CSF enhances tumor invasion by elevated MMP-2, -9, and -26 expression. Cancer Med. 2013 Apr;2(2):117-29. doi: 10.1002/cam4.20. Epub 2012 Nov 26. PMID: 23634280;PMCID: PMC3639651.
9: Oshika Y, Nakamura M, Abe Y, Fukuchi Y, Yoshimura M, Itoh M, Ohnishi Y,
Tokunaga T, Fukushima Y, Hatanaka H, Kijima H, Yamazaki H, Tamaoki N, Ueyama Y. Growth stimulation of non-small cell lung cancer xenografts by granulocyte-macrophage colony-stimulating factor (GM-CSF). Eur J Cancer. 1998Nov;34(12):1958-61. doi: 10.1016/s0959-8049(98)00236-6. PMID: 10023322.
10: Reckamp KL, Gardner BK, Figlin RA, Elashoff D, Krysan K, Dohadwala M, Mao J, Sharma S, Inge L, Rajasekaran A, Dubinett SM. Tumor response to combination celecoxib and erlotinib therapy in non-small cell lung cancer is associated with a low baseline matrix metalloproteinase-9 and a decline in serum-soluble E-cadherin. J Thorac Oncol. 2008 Feb;3(2):117-24. doi:
10.1097/JTO.0b013e3181622bef. PMID: 18303430.
11: Reckamp KL, Koczywas M, Cristea MC, Dowell JE, Wang HJ, Gardner BK, Milne GL, Figlin RA, Fishbein MC, Elashoff RM, Dubinett SM. Randomized phase 2 trial of erlotinib in combination with high-dose celecoxib or placebo in patients with advanced non-small cell lung cancer. Cancer. 2015 Sep 15;121(18):3298-306. doi:10.1002/cncr.29480. Epub 2015 May 29. PMID: 26033830; PMCID: PMC4864011.
12: Edelman MJ, Wang X, Hodgson L, Cheney RT, Baggstrom MQ, Thomas SP, Gajra A, Bertino E, Reckamp KL, Molina J, Schiller JH, Mitchell-Richards K, Friedman PN, Ritter J, Milne G, Hahn OM, Stinchcombe TE, Vokes EE; Alliance for Clinical Trials in Oncology. Phase III Randomized, Placebo-Controlled, Double-Blind Trial of Celecoxib in Addition to Standard Chemotherapy for Advanced Non-Small-Cell Lung Cancer With Cyclooxygenase-2 Overexpression: CALGB 30801 (Alliance). J Clin
Oncol. 2017 Jul 1;35(19):2184-2192. doi: 10.1200/JCO.2016.71.3743. Epub 2017 May10. PMID: 28489511; PMCID: PMC5493050.
13: Halatsch ME, Kast RE, Karpel-Massler G, Mayer B, Zolk O, Schmitz B, Scheuerle A, Maier L, Bullinger L, Mayer-Steinacker R, Schmidt C, Zeiler K, Elshaer Z, Panther P, Schmelzle B, Hallmen A, Dwucet A, Siegelin MD, Westhoff MA, Beckers K, Bouche G, Heiland T. A phase Ib/IIa trial of 9 repurposed drugs combined with temozolomide for the treatment of recurrent glioblastoma: CUSP9v3. Neurooncol Adv. 2021 Jun 24;3(1):vdab075. doi: 10.1093/noajnl/vdab075. PMID: 34377985;
PMCID: PMC8349180.
14: Gupta R, Cristea M, Frankel P, Ruel C, Chen C, Wang Y, Morgan R, Leong L, Chow W, Koczywas M, Koehler S, Lim D, Luu T, Martel C, McNamara M, Somlo G, Twardowski P, Yen Y, Idorenyi A, Raechelle T, Carroll M, Chung V. Randomized trial of oral cyclophosphamide versus oral cyclophosphamide with celecoxib for recurrent epithelial ovarian, fallopian tube, and primary peritoneal cancer. Cancer Treat Res Commun. 2019;21:100155. doi: 10.1016/j.ctarc.2019.100155. Epub2019 Jul 3. PMID: 31279962; PMCID: PMC9018111.
15: Chen EY, Blanke CD, Haller DG, Benson AB, Dragovich T, Lenz HJ, Robles C, Li H, Mori M, Mattek N, Sanborn RE, Lopez CD. A Phase II Study of Celecoxib With Irinotecan, 5-Fluorouracil, and Leucovorin in Patients With Previously Untreated Advanced or Metastatic Colorectal Cancer. Am J Clin Oncol. 2018 Dec;41(12):1193-1198. doi:10.1097/COC.0000000000000465. PMID: 29782360; PMCID:PMC6240505.
16: Gulyas M, Mattsson JSM, Lindgren A, Ek L, Lamberg Lundström K, Behndig A, Holmberg E, Micke P, Bergman B; Swedish Lung Cancer Study Group. COX-2 expression and effects of celecoxib in addition to standard chemotherapy in advanced non-small cell lung cancer. Acta Oncol. 2018 Feb;57(2):244-250. doi:10.1080/0284186X.2017.1400685. Epub 2017 Nov 15. Erratum in: Acta Oncol. 2018Apr;57(4):564. PMID: 29140138.
17: Takhar H, Singhal N, Mislang A, Kumar R, Kim L, Selva-Nayagam S, Pittman K, Karapetis C, Borg M, Olver IN, Brown MP. Phase II study of celecoxib with docetaxel chemoradiotherapy followed by consolidation chemotherapy docetaxel plus cisplatin with maintenance celecoxib in inoperable stage III nonsmall cell lung cancer. Asia Pac J Clin Oncol. 2018 Feb;14(1):91-100. doi:10.1111/ajco.12749. Epub 2017 Aug 25. PMID: 28840978.
Author Response
Please see the attachment.

Reviewer 3 Report (Previous Reviewer 3)
The authors made reasonable adjustments and the IHC data confirms their earlier conclusions which strengthens them.
There are a few text errors that should be adjusted, but this is easily adjusted by going over the text again.
Author Response
Please see the attachment.

Reviewer 4 Report (Previous Reviewer 4)
Comments have been addressed.
Author Response
Please see the attachment.

This manuscript is a resubmission of an earlier submission. The following is a list of the peer review reports and author responses from that submission.
Round 1
Reviewer 1 Report
The authors' work adds new insights into the natural history of vestibular schwannoma.
However, it has some issues to be addressed on the other hand.
[Lines 79-82]
The format in which the study results are described in the Introduction is uncommon. Therefore, this section should conclude with the aims of this study.
[Lines 84-104]
The authors' study excluded a large number of patients and was strongly influenced by selection bias, which should be clearly stated as a limitation of the study.
[Tables 2 and 3]
Tables 2 and 3 show the mean and range of each value.
The mean value may be strongly influenced by the minimum or maximum value in a relatively small number of cohorts, such as in the authors' study, and updating the median value is strongly recommended.
[Lines-228-336]
Differences in the natural history of vestibular schwannomas have been reported depending on the tumor's location, such as the internal auditory canal and those with CP angle extension; have the authors considered this point? This point should be analyzed.
Reviewer 2 Report
This paper confirms and consolidates elements of previous findings on VS with potentially great importance to development of new treatments. It is a rare paper indeed that sheds such core data on a common serious disease as does this paper. I heartily support publication of this work. The main matter that must be addressed [fixed or explain my misunderstanding in the text] before publication is marked ### below.
Although grammar and medical English use are generally good, the scattered language use errors, disorganized, jumbled exposition, and odd, unclear constructions indicate that this paper has not been professionally edited. The Introduction must be rewritten taking care to have one main subject within a paragraph. As the paper stands the authors go back-and-forth between subjects within the same run-on paragraph. This is sad because the paper presents a major advance in our understanding VS, assuming the authors’ findings are confirmed.
If my understanding is correct, the authors should clearly state in the Conclusion section [or in Discussion section] that they have consolidated previous findings on CD68 and CD163 in VS.
The Discussion section must be rewritten. F.ex. the paragraph 288 to 293 is empty verbiage. It adds nothing. A discussion should briefly bring up ancillary thoughts, connections, conjectures, and consequences coming from both the positive and negative findings of the paper. It is in that context that very short mention of findings already presented in the Results section can be restated. But in no case can simple recounting of what was studied be acceptable. Starting the Discussion section with line 294, “Our results showed…” would be fine.
Given the puzzling findings of no increase in Ki-67 in faster growing VS, the value of this paper would be increased by oil immersion, x 100 H&E micrographs of low vs. fast growing VS. If I were the Editor I would require this. Given the authors' findings the usual x 400 would not be helpful. Would the first thought be VS volume increase being constituted by infiltrating monocytic lineage cells ?
Line 15, “...schwannomas makes therapy prediction very…“. No. It makes therapy decisions difficult or makes prognosis prediction difficult, or makes therapy timing prediction difficult, etc.
L 25, an interesting point is why VS are so common, where other cranial n. S are less common.
L 34, Ki-67 is not a growth factor. It is a nuclear protein.
L 50, re. “contrasted Magnetic Resonance Imaging (MRI)” I dont think MRI needs explaining. Better would be “contrast MRI”
L51, “suffering from” is a bit archaic. Better is simply “with”
L 63, “In recent years…” start a new paragraph. Also in a separate paragraph a few sentences on CD68 and a few sentences on CD163 is needed. These are of particular importance to the authors findings
L 67, The Vascular Endothelial…
L 69, if the subject of M2 is brought up, it must be done in its own paragraph with a quick overview of M1, M2 and mention of unclear elements that are well known to those who study macrophage-related cells but are not well known to general medical and neurosurgeons or ENT surgeons who will all be reading this (hopefully).
L 72, Ki-67 is well enough known as to not need prior explanation. Mentioning Kiel adds confusion.
Re. reference to Koos scale, the authors could consider adding a brief mention, maybe in a Table legend, of Koos 1= VS restricted to intracanalicular. 2= < 2cm extracanalicular extension, 3= cerebellopontine angle VS <3m and not displacing cerebellar trunk, 4= VS >4 cm with brainstem displacement. This would help non-specialist GPs.
Ll 294, “ Our results showed…” makes no sense to me.
Line 302, Error. ASA is a dual COX1-2 inhibitor.
Line 306 should start a new paragraph.
Line 313. Numerous problems here require fixing. 1] the data of Reckamp et al [below] must be discussed in context of any clinical study of celecoxib. 2] Since the authors’ study was on ex vivo IHC tissue the authors cannot state that there results are discrepant with others’ data showing COX-2 inhibition inhibited growth can they ? 3] The growth inhibition literature largely was done in cancer. Malignant tissue may [probably] has different growth drives than does non malignant growth. VS is a non malignant growth. Agreed ?
Line 336. “Inflammatory infiltrate” is not good enough for those of us on the hunt for better treatments. Which inflammation related cells ? drawn to tumor tissue by which chemokines ? those chemokines’ synthesis are triggered by what signals ? etc. The authors work would have been much improved by adding IHC for neutrophils, lymphocytes, mast cells, etc.
Would the authors consider it useful adding a table with the differences between M1, M2, and Mnul macrophages ?
The discrepancy between clearly increased expression of CD68 in fast growing VS versus the non-meaningful difference in CD163 needs discussing. Can the authors account for this ?
###A major flaw in this paper is the lack of M-CSF measurement details [I might have missed this ?]. Did the authors measure serum or IHC M-CSF ? If it was serum then controls’ M-CSF must be measured with the same method as used in VS pts. I do not consider the difference between slow and fast growing VS levels of M-CSF to be meaningful. Also the slight differences in M-CSF would not seem enough to account for the differences in CD68. In malignant tumors generally it is more usual for GM-CSF to drive macrophage infiltration. This paper would have been much improved if this also was tested.
The authors could argue against my view - which might be idiosyncratic or wrong - that note should be made that statistical differences are not always meaningful differences. F. ex. I see the only meaningful differences between slow and fast growing VS are greater CD68, VS volume, and hearing loss in fast growing VS. If the authors agree with this, they should say that clearly and simply in the Abstract and in Conclusion sections. If they don’t, could the authors state why the small differences in other parameters meet not only statistical significance but also meet clinically meaningful criteria ? Then also state what the clinically meaningful consequence might be ?
==============================
Reckamp KL, Krysan K, Morrow JD, Milne GL, Newman RA, Tucker C, Elashoff RM, Dubinett SM, Figlin RA. A phase I trial to determine the optimal biological dose of celecoxib when combined with erlotinib in advanced non-small cell lung cancer. Clin Cancer Res. 2006 Jun 1;12(11 Pt 1):3381-8. doi: 10.1158/1078-0432.CCR-06-0112. PMID: 16740761.
Reviewer 3 Report
the authors present an interesting study on six markers Ki-67, COX2, VEGF, M-CSF, CD163, and CD68 on vestibular schwannoma (VS) progression and tumor size in a cohort of 171 VS. The markers are determined by qPCR.
Although the authors perform extended analysis and the data is well presented, the conclusions they draw based solely on qPCR data seems a bit far fetched. As they correctly reference a number of papers that look into these markers by immunohistochemistry, it does not seem that this data adds much to the excisiting literature.
More in depth analysis of their data, for instance by looking with immunohistochemistry to see if the CD 163 and CD68 cells are actual macrophages, would be recommended before drawing these conclusions.
Reviewer 4 Report
This is an interesting and important study, for variability in growth rate of VS is a challenging clinical problem. The authors are to be commended for undertaking a comprehensive study.
However, the findings are in contrast to many other studies. Why is this the case? The authors should consider and discuss this.
Furthermore, the increase in macrophage markers does not mean that the increase in volume of these tumors is due to an increase in the number of macrophages rather than tumor cells. This could be due to immune signaling that produces an increase in tumor cells via a downstream pathway. This conclusion needs modification.
Immunohistochemical studies of 10-20 high power fields of each tumor could sort this out further. But as stated, the conclusion cannot be supported.